# Sarcopenia in Breast Cancer Patients: A Systematic Review and Meta-Analysis

**DOI:** 10.3390/cancers16030596

**Published:** 2024-01-31

**Authors:** Michela Roberto, Giacomo Barchiesi, Blerina Resuli, Monica Verrico, Iolanda Speranza, Leonardo Cristofani, Federica Pediconi, Federica Tomao, Andrea Botticelli, Daniele Santini

**Affiliations:** 1UOC Oncologia A, Department of Hematology, Oncology and Dermatology, Policlinico Umberto I University Hospital, Sapienza University o f Rome, Viale Regina Elena, 324, 00161 Rome, Italy; michela.roberto@uniroma1.it (M.R.); g.barchiesi@policlinicoumberto1.it (G.B.); monica.verrico@uniroma1.it (M.V.); iolanda.speranza@uniroma1.it (I.S.); leonardo.cristofani@uniroma1.it (L.C.); andrea.botticelli@uniroma1.it (A.B.); daniele.santini@uniroma1.it (D.S.); 2Department of Medicine V, University Hospital Munich, Ziemssenstraße 5, 80336 Munich, Germany; 3Department of Radiological, Oncological and Pathological Sciences, Sapienza University of Rome, Viale Regina Elena, 324, 00161 Rome, Italy; federica.pediconi@uniroma1.it; 4Department of Maternal and Child Health and Urological Sciences, Sapienza University of Rome, Viale Regina Elena, 324, 00161 Rome, Italy; federica.tomao@uniroma1.it

**Keywords:** sarcopenia, breast cancer, lean muscle mass, skeletal muscle index (SMI), prognostic factor, body composition

## Abstract

**Simple Summary:**

The evaluation of additional body composition measures, such as visceral adipose tissue area, subcutaneous adipose tissue area, and sarcopenic obesity, could be useful to improve our understanding of the prognostic role of body composition parameters in women with breast cancer. The aim of our review was to summarize current evidence about sarcopenia in non-metastatic as well as metastatic breast cancer (MBC) and to identify any correlation between sarcopenia and patient outcomes. We observed a correlation between sarcopenia and significantly higher rates of treatment-related toxicities in both settings (metastatic and non-metastatic) compared with patients without sarcopenia. It was found that sarcopenic patients were more likely to deal with severe toxicities compared to patients classified as non-sarcopenic. This finding suggests that sarcopenia recently emerged as a new condition that, independently from malnutrition, may adversely affect patient outcomes and may be used as a reference for chemotherapy dose selection to better balance individual pharmacokinetic differences.

**Abstract:**

**(1) Background:** We estimated the prevalence and clinical outcomes of sarcopenia among breast cancer patients. **(2) Methods:** A systematic literature search was carried out for the period between July 2023 and October 2023. Studies with breast cancer patients evaluated for sarcopenia in relation to overall survival (OS), progression-free survival (PFS), relapse of disease (DFS), pathological complete response (pCR), or toxicity to chemotherapy were included. **(3) Results:** Out of 359 screened studies, 16 were eligible for meta-analysis, including 6130 patients, of whom 5284 with non-MBC. Sarcopenia was evaluated with the computed tomography (CT) scan skeletal muscle index and, in two studies, with the dual-energy x-ray absorptiometry (DEXA) appendicular lean mass index. Using different classifications and cut-off points, overall, there were 2007 sarcopenic patients (33%), of whom 1901 (95%) presented with non-MBC. Sarcopenia was associated with a 33% and 29% higher risk of mortality and progression/relapse of disease, respectively. Sarcopenic patients were more likely to develop grade 3–4 toxicity (OR 3.58, 95% CI 2.11–6.06, *p* < 0.0001). In the neoadjuvant setting, a higher rate of pCR was observed among sarcopenic patients (49%) (OR 2.74, 95% CI 0.92–8.22). **(4) Conclusions:** Our meta-analysis confirms the correlation between sarcopenia and negative outcomes, especially in terms of higher toxicity.

## 1. Background

Breast cancer is the most common cancer and the second leading cause of cancer deaths among women [1]. The incidence did not decline despite decades of widespread use of population-based mammography screening [1]. Approximately 5% of all cases present with metastatic disease at diagnosis, while most patients develop distant metastases after curative treatments for localized disease [2]. The majority of breast cancer is diagnosed at an early stage (I–III), and 20–30% will eventually develop metastases [2]. Metastases are most commonly identified months to years after initial breast cancer diagnoses described as recurrent. Some patients present with distant sites of disease at initial diagnosis, termed de novo MBC [2].

Body composition parameters recently became a field of great interest in cancer research, with growing evidence that they can correlate with cancer prognosis. Specifically, there is an emergent recognition that body mass index (BMI, weight in kilograms divided by the square of height in meters) is not adequate to identify patients who are at risk of adverse outcomes.

Recently, sarcopenia emerged as a crucial factor for survival and treatment-related complications in patients with solid tumors. Sarcopenia can be defined as the progressive degeneration of muscle mass and strength, as well as decreased physical activities in older adults [3]. Several factors were related to the sarcopenia outbreak: inflammation, declined nutrition intake, and neurodegeneration. In cancer patients, malnutrition and chronic inflammation can also impair muscle tissue by promoting muscle loss. Sarcopenia, considered as the lower percentage of lean muscle mass, can be estimated by a CT scan, magnetic resonance imaging (MRI), DEXA, or bioelectric impedance analysis (BIA) [4]. It also represents a hallmark of cachexia, a complex syndrome characterized by unintended loss of both tissue and lean body mass [5]. Cancer cachexia is a condition of tissue wasting which develops as a secondary disorder in cancer patients and leads to progressive functional impairment [5]. It is characterized by systemic inflammation; negative protein and energy balance; and involuntary loss of lean body mass, with or without wasting of adipose tissue [6]. The onset of cachexia is strictly connected to lower adherence to cancer treatments [6] and to an increased likelihood of infections and reduced mobility.

Previous systematic reviews and meta-analyses showed that sarcopenia correlates with poorer survival in digestive tract cancer (e.g., esophageal, gastric, pancreatic, and colorectal cancer) [7,8,9,10] and lung cancer [11]. According to the literature, sarcopenia is a poor prognostic indicator in oncology and an independent predictor of the occurrence of severe postoperative complications in different solid tumors [7,8,9,10].

More recently, body composition was evaluated in several studies on breast cancer, and an artificial intelligence (AI) model that can predict five-year survival in patients with stage IV metastatic breast cancer included sarcopenia among the selected prognostic factors [12]. Most of the published results showed that there is a significant association between sarcopenia and a higher risk of mortality [13,14,15,16,17,18,19] and toxicity during chemotherapy in breast cancer patients. These studies demonstrated also that sarcopenia was associated with worse global and physical functioning of health-related quality of life scores and may reflect a multifactorial and bidirectional relationship between skeletal muscle status and wellbeing [20,21]. The role of reduced skeletal muscle mass in the decline of physical strength is well established, and there is evidence of a link between strength and health-related quality of life in cancer patients. Furthermore, cancer patients who experience severe toxicities have a worse quality of life and physical function overall. Athough other studies reported conflicting results [22,23,24,25,26], sarcopenia resulted to be correlated with higher rates of toxicity [27,28,29,30,31,32], thus we believe that it could indirectly influence the quality of life of cancer patients. The published results in these studies are based on relatively small numbers of patients.

In this systematic review and meta-analysis, we aim to summarize current evidence about sarcopenia in both non-metastatic as well as metastatic breast cancer (MBC) and to identify any correlation between sarcopenia and patient outcomes, especially in terms of toxicity and survival.

## 2. Material and Methods

### 2.1. Search Strategy

A systematic search of the literature for randomized controlled trials and observational studies was conducted according to the Preferred Reporting Items for Systematic reviews and Meta-Analysis (PRISMA) guidelines. The literature search on Pubmed-MEDLINE was performed between July 2023 and October 2023. The searching strategy was as follows: ((breast[Title/Abstract]) AND (cancer[Title/Abstract] OR tumor[Title/Abstract] OR neoplasms[Title/Abstract] OR carcinoma[Title/Abstract])) AND ((sarcopenia*[Title/Abstract]) OR (sarcopenic[Title/Abstract])) OR (sarcopenia[MeSH Terms])) OR (sarcopenia[Title/Abstract])). Keywords “sarcopenia, breast cancer, survival, toxicity” were used for searching in Cochrane Central Register of Controlled Trials and SCOPUS. No poster presentation or abstract was included. The systematic review followed the recommendations of the Preferred Reporting Items for Systematic Reviews and Meta-Analysis (PRISMA). The protocol was not registered.

### 2.2. Study Selection Criteria and Data Extraction

Studies with breast cancer patients (men or women) evaluated for sarcopenia, regardless of treatment or disease stage, in relation to overall survival (OS), progression-free survival (PFS), disease-free survival (DFS), breast cancer-specific survival (BCSS), pathological complete response (pCR), or safety profile related to chemotherapy, were included in our analysis. We limited the review to prospective clinical trials (if available) and retrospective or prospective cohort series including more than 10 patients. Case reports and case series, editorials, commentaries, meta-analyses, review articles, and animal studies were excluded. Concerning the selection criteria, only studies written in the English language were included in the meta-analysis.

Only studies where sarcopenia was defined according to CT scan radiological criteria and expressed as skeletal muscle index (SMI) or as appendicular lean mass index (LMI) according to DXA scan were included. Studies that evaluated sarcopenia as a continuous variable, or by BIA or MRI, were not included in the present analysis. Moreover, sarcopenia described as pectoralis muscle area (PMA) depletion was not included. Among the included studies, sarcopenia was diagnosed at BC diagnosis or at the time of starting therapy. The primary endpoint was to evaluate the correlation between sarcopenia and survival outcomes: OS, defined as the time from sarcopenia diagnosis until death from any cause; PFS, defined as the time from sarcopenia diagnosis until progression of disease or death from any cause; DFS, defined as the time from sarcopenia diagnosis until disease relapse in the early stage; and BCSS, defined as the time from sarcopenia diagnosis until death from breast cancer. Secondary endpoints were the correlation between sarcopenia and (a) grade > 2 toxicities, according to Common Terminology Criteria for Adverse Events (CTCAE 4.0); (b) pCR, defined as absence of invasive/in situ cancer in the breast and/or axillary lymph nodes after neoadjuvant chemotherapy.

Data were extracted independently by 2 authors (B.R. and M.V.) and entered into a standardized, predesigned Microsoft Excel form. The following data were recorded: author, publication year, and study design; median time of follow-up (if reported); number of total patients; median age of patients; setting (i.e., neoadjuvant, non-metastatic, and metastatic) and hisotypes (if specified) of breast cancer disease; kind of treatment (chemotherapy, endocrine-therapy, radiotherapy); the definition of sarcopenia (SMI, varied cut points); and study outcomes (i.e., OS, PFS, DFS, BCSS, toxicity, or pCR). Each author assessed the quality of reporting.

### 2.3. Statistical Analysis

In both non-metastatic and metastatic settings, a random-effect meta-analysis model was applied, and the inverse-variance weighting was used to pool estimates of the included studies. The meta-analysis was conducted by Rev-Man 5.4 software. The importance of the observed value of I^2^ depends on the magnitude and direction of effects and the strength of evidence for heterogeneity. Statistical significance was defined at the 0.05 level. Dichotomous data were used to assess inverse variance and risk ratio (RR) for toxicity, pCR, and BCSS. Generic inverse variance was expressed in log hazard ratios (HR) with 95% confidence intervals (CI) for OS, PFS/DFS, and BCSS.

## 3. Results

### 3.1. Literature Search and Included Studies

Overall, 359 records were identified using the search strategy, and we excluded 188 duplicate articles. After screening the titles and abstracts, 138 records were excluded based on the type of article (review, case report, clinical trials) or irrelevant content. Of the remaining 35 potentially relevant studies, we excluded 16 papers for not having enough data available or for not meeting our inclusion criteria. Finally, we selected 16 articles published worldwide between 2009 and 2023, among which there were six prospective studies. The selection process is shown in Figure 1.

### 3.2. Patient and Study Characteristics

We analyzed 16 articles involving patients with non-metastatic (N = 10, with three studies in neoadjuvant settings) or MBC (N = 5) and one study including both settings. The median follow-up range throughout the studies is from 6 months (in the earlier studies) to >6 years (in the older one). A total of 6130 patients, of which 5284 presented with non-MBC, ranging in age from 18 to 89 years, were included in the present meta-analysis. Patients were treated with surgery, radiotherapy, endocrine therapy, chemotherapy, or targeted therapy, according to the histotype and stage of the disease. In all included studies, sarcopenia was defined by a lower SMI than normal patients, even with different cut-offs, expressed in cm^2^/m^2^ (SMI ≤ 41 in five studies, SMI ≤ 40 in four studies, SMI ≤ 38.5 in three studies, and others cut-off in single studies) while two studies reported sarcopenia as LMI < 5.45 Kg/m^2^ (Table 1). Overall, there were 2007 sarcopenic patients (33%), of whom 1901 (95%) presented with non-MBC.

We also described breast cancer subtypes if they were reported in the studies. All other main characteristics are listed in Table 1.

### 3.3. Outcomes

#### 3.3.1. Survival Analysis

For OS analysis, we included nine studies, of which Shachar and Ballinger’s [15,25] were conducted in the MBC setting, and Palleschi et al. [24] considered both non-metastatic and MBC settings together. In meta-analysis, sarcopenia was associated with a 33% greater mortality risk compared to non-sarcopenic patients (HR 1.33, 95%CI 0.97–1.80, *p* = 0.07, I^2^ = 71%) (Figure 2). No difference was shown by the subgroup analysis according to the disease setting (Figure 2). In terms of PFS/DFS, sarcopenic patients reported a 29% greater risk of progression/relapse of disease compared to non-sarcopenic patients (four studies, HR 1.29, 95% CI 0.79–2.10, *p* = 0.32, I^2^ = 50%) (Figure 3). Considering the BCSS, the risk of mortality from breast cancer was higher in sarcopenic patients with non-MBC (two studies, HR 1.51, 95% CI 0.86–2.63, *p* = 0.15, I^2^ = 0%) (Figure 4).

#### 3.3.2. Response

The study of Del Fabbro et al. explored the correlation between sarcopenia and the pCR in the neoadjuvant setting [26]. The study enrolled 129 patients (HER2+ = 44, ER+ = 33, clinical stage III = 28), of whom 18 (14%) presented with sarcopenic conditions (Table 1). Patients with sarcopenia (72%) reported a higher rate of pCR compared to the control arm (49%) (OR 2.74, 95% CI 0.92–8.22), even though it was not statistically significant at univariate analysis (*p* = 0.07).

#### 3.3.3. Toxicity

Six studies reported data on grade >2 toxicity. Among them, three (Shachar 2017, Prado 2009, and Deluche 2022 [15,21,31]) were conducted in MBC settings. Patients with sarcopenia (42%) had more grade 3–4 toxicity compared to patients classified as non-sarcopenic (19%) (OR 3.58 95% CI 2.11–6.06) (*p* < 0.0001) (I^2^ = 29%) (Figure 5).

## 4. Discussion

There is a rapidly growing interest in exploring the impact of sarcopenia on toxicity and outcomes in early and advanced breast cancer. Previously published meta-analyses [33,34] investigated survival outcomes and toxicity from oncological therapies in sarcopenic patients with breast cancer.

Our results appeared to be particularly relevant when considering the correlation between sarcopenia and toxicity. We found a higher rate of toxicity from oncological therapies in sarcopenic patients in both metastatic and non-metastatic breast patients, and especially more grade 3–4 toxicity in MBC. Our results are in line with findings reported in a previous meta-analysis [33] showing more grade 3–5 toxicities in sarcopenic patients compared to non-sarcopenic patients (42% vs. 19%; OR 3.58 95% CI 2.11–6.06, 6 studies; *p* < 0001; I^2^ = 29%). The only study included in our review that did not show any correlation between sarcopenia and toxicity was the study from Deluche 2022 et al. [31]. Adverse events (AEs) overall occurred in 8.6% of patients included in the study from Deluche et al., compared to >20% of patients in the other studies. The authors concluded their study was a cross-sectional design, in contrast to other studies that were especially designed to evaluate the relationship between toxicities and sarcopenia. Another explanation could be the use of treatments other than chemotherapy or the use of less toxic treatments rather than those used in previous studies.

Our results provide useful information because they demonstrate that sarcopenic patients are more vulnerable to the side effects of chemotherapy. Hence, sarcopenia screening should be included in dose calculation for chemotherapy, especially in MBC patients. Moreover, this finding suggests that sarcopenia recently emerged as a new condition that, independently from malnutrition, may adversely affect patient outcomes and may be used as a reference for chemotherapy dose selection to better balance individual pharmacokinetic differences. The use of body composition measurements to individualize dosing could represent a dramatic step forward in the personalized medicine era.

Furthermore, we found that breast cancer patients with sarcopenia have a 33% increased mortality risk and a 29% increased risk of progression/relapse of the disease compared to non-sarcopenic patients. Previous studies reported the negative correlation between sarcopenia and survival outcomes in several tumors, including breast cancer, confirming the prognostic role of sarcopenia in cancer patients [8,10]. Our meta-analysis confirms the negative correlation between sarcopenia and OS observed in the works of Xiao-Ming and Aleixo. According to subgroups, sarcopenia did not increase mortality in patients with MBC. Interestingly, the subgroup analysis in the study of Aleixo et al. [33] showed that sarcopenia was not always significant in early breast cancer compared to MBC. In this study, low muscle density was prognostic for OS in MBC, but not in the early stage. Unfortunately, we could not perform subgroups by stage because only one trial exclusively enrolled patients with MBC.

There is growing interest in identifying the mechanisms that may explain the strong correlation between sarcopenia and mortality. Several factors were proposed; muscles represent an energy “reservoir” which can be exploited during catabolic periods such as cancer and or chemotherapy [35]. Commonly, all cancer patients are subjected to degenerative factors such as aging, malnutrition, and physical inactivity, which are all potent causes of muscle dysfunction. Patients with limited physical reserve or individual vulnerability to cancer or cancer treatments may exhaust their energy storages, enhancing sarcopenia’s impact on prognosis [36]. In addition, sarcopenia is primarily characterized by muscle loss, which is often the result of an imbalance between protein synthesis and degradation. Such imbalance increases cell apoptosis, reducing regenerative capacity [37]. Finally, sarcopenia is related to a condition of systemic inflammation that occurs both at pre-clinical and clinical levels: it contributes to activate tumor necrosis factor (TNF) cascade [38] and is linked to a high neutrophil to lymphocyte ratio [39]. TNF cascade promotes tumor invasion and migration [40], while a high neutrophil-to-lymphocyte ratio increases mortality [41].

Interestingly, according to the study of Del Fabbro et al., there was a positive trend between sarcopenia and pCR in BC patients treated with neoadjuvant chemotherapy [26]. This is unexpected because previous published studies showed worse outcomes for sarcopenic patients [41]. However, the results must be interpreted with caution. Firstly, a small number of patients were included in the analysis. Secondly, despite a trend toward a higher rate of pCR in sarcopenic patients, the response to neoadjuvant chemotherapy was not significantly associated with sarcopenic status. Finally, it was hypothesized that patients with early breast cancer are somehow healthier than patients included in previous reports (especially those with metastatic tumors). This factor may have positively influenced outcomes; patients were able to receive a higher chemotherapy dosage but were also able to better tolerate chemotherapy toxicities than patients with advanced tumors. Moreover, they were less likely to be affected by sarcopenia because of cancer cachexia syndrome, which is more typical in other metastatic cancers than in BC [42].

Several ongoing studies are evaluating strategies to prevent or revert sarcopenic status in BC patients [43,44]. One of the most promising approaches is represented by physical activity. Physical activity is now recognized as an important therapeutic intervention to maintain bone integrity in postmenopausal women treated with estrogen suppressors [43]. Preclinical studies showed that physical exercise enhances crosstalk between skeletal and muscular systems. The exploitation of this crosstalk has great potential to avoid musculoskeletal complications of sarcopenia [45,46]. A nutritional counseling and protein-based intervention to treat sarcopenic status and improve lean mass percentage in body composition is also being evaluated [44]. This is fundamental when considering that vegetable protein intake was associated with statistically significant lower breast cancer incidence and statistically lower deaths after breast cancer diagnosis, whereas higher animal protein intake was associated with statistically higher breast cancer incidence.

For most patients, the focus should be on identifying those at increased risk of malnutrition and/or muscle depletion at the time of diagnosis since this situation could be related to a sarcopenia condition at the onset of cancer diagnosis [47] and during treatment [48]. Muscaritoli et al. deployed a nutrition awareness protocol (PRONTO), a standardized approach for the identification and monitoring of nutritional risk of patients commencing and undergoing antineoplastic therapy [48]. This protocol enables the rapid identification of patients with, or at risk of, malnutrition and or muscle depletion. This protocol is adjustable to all settings of patients and countries, and therefore, the application is feasible by oncologists in their daily care in order to improve patient outcomes. The association between malnutrition and sarcopenia is also highlighted in the NUTRIONCO study observations of Muscaritoli et al., in which a statistically significant association was reported between overall survival and the baseline nutritional status of cancer patients, revealing that malnutrition decreased survival probability, mainly in non-metastatic patients [49].

Our study has some limitations. First, we included non-randomized clinical trials, which contain confounding factors that may have affected our results. Second, few studies were included in the meta-analysis, which limited the chance to perform subgroup analysis. Furthermore, the definition of sarcopenia was not uniform across studies, limiting comparison and the standardization of the process. Finally, any difference based on cancer subtypes or the kind of treatment cannot be established because the outcome for sarcopenic patients was not analyzed for these parameters in the studies.

## 5. Conclusions

Our results show that body composition is correlated with cancer and toxicity outcomes and suggest that breast cancer patients with low muscle mass in both metastatic and not-metastatic settings have more treatment-related toxicity. Therefore, sarcopenia can be used to predict treatment-related toxicity and should be considered a robust prognostic factor of negative healthcare costs. Using baseline CT scan imaging and readily available software, skeletal muscle mass assessments could be incorporated into the clinical setting, and we hypothesize that they could prevent severe adverse, dose-limiting events. Further research is needed to investigate the prognostic role of sarcopenia in breast cancer patients, as well as to understand the deeper mechanisms underlying the sarcopenia status, systemic inflammation, and outcomes in cancer patients. Further research exploring the role of body composition in pharmacokinetics is needed, with a focus on alternative dosing strategies in sarcopenic patients.

## Figures and Tables

**Figure 1 cancers-16-00596-f001:**
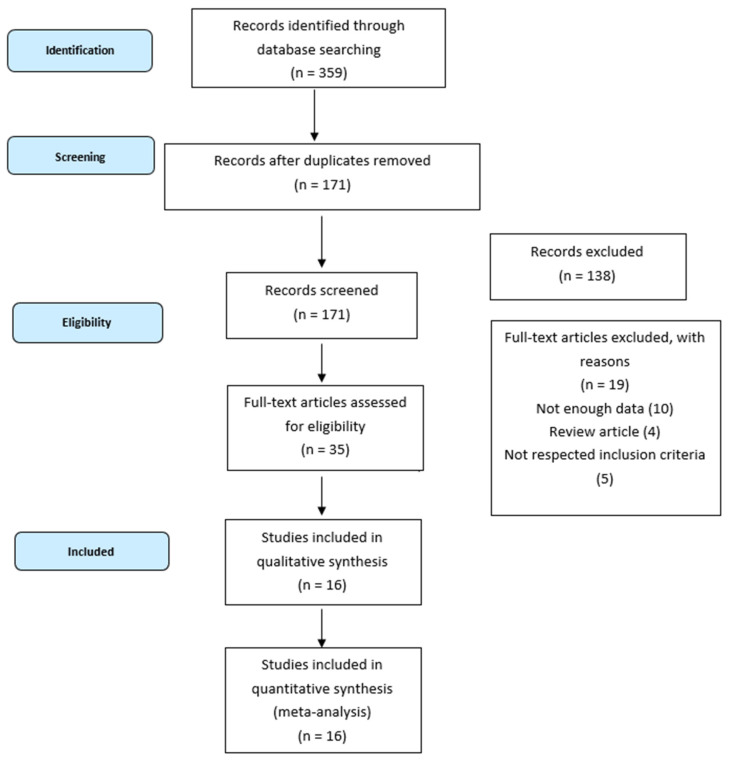
Flowchart of the selection process.

**Figure 2 cancers-16-00596-f002:**
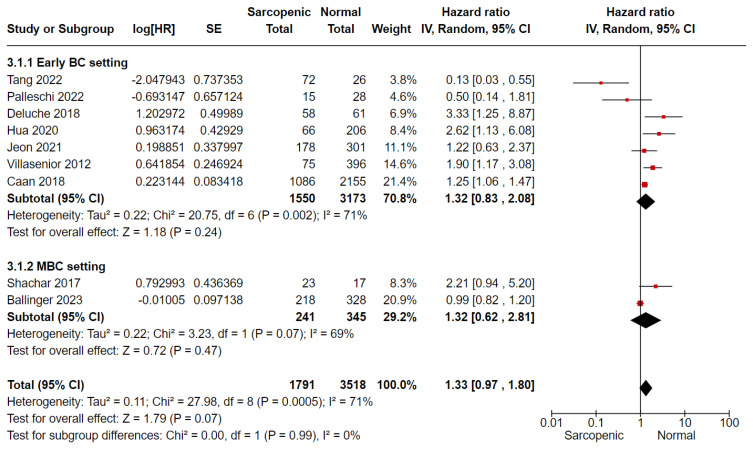
Forest plot analysis for overall survival and sarcopenia in breast cancer patients, according to disease setting, Tang et al. [23], Pallerschi 2022 et al. [24], Deluche 2018 et al. [17], Hua 2020 et al. [18], Jeon 2021 et al. [22], Villasenoir 2012 et al. [14], Caan 2018 et al. [16], Shachar 2017 et al. [15], Ballinger 2023 et al. [25].

**Figure 3 cancers-16-00596-f003:**
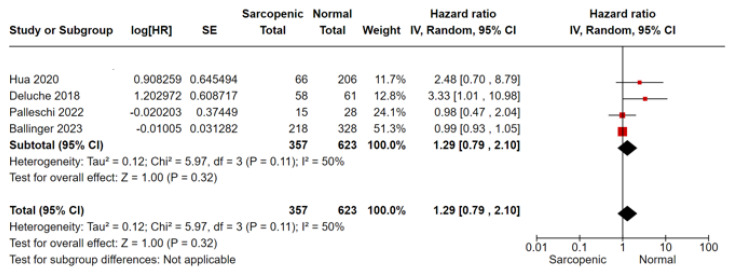
Forest plot for progression/disease-free survival and sarcopenia in breast cancer patients, Hua 2020 et al. [20], Deluche 2018 et al. [17], Palleschi 2022 et al. [24], Ballinger 2023 et al. [25].

**Figure 4 cancers-16-00596-f004:**
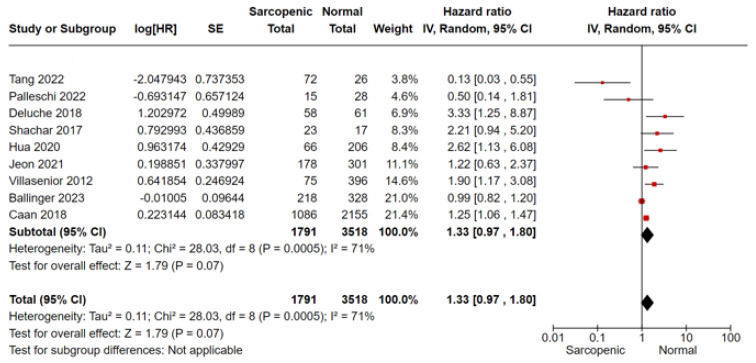
Forest plot analysis for breast cancer-specific survival and sarcopenia in breast cancer patients, Tang 2022 et al. [23], Palleschi 2022 et al. [24], Deluche 2018 et al. [17], Shachar 2017 et al. [15], Hua 2020 et al. [18], Jeon 2021 et al. [22], Villasenoir 2012 et al. [14], Ballinger 2023 et al. [25], Caan 2018 et al. [16].

**Figure 5 cancers-16-00596-f005:**
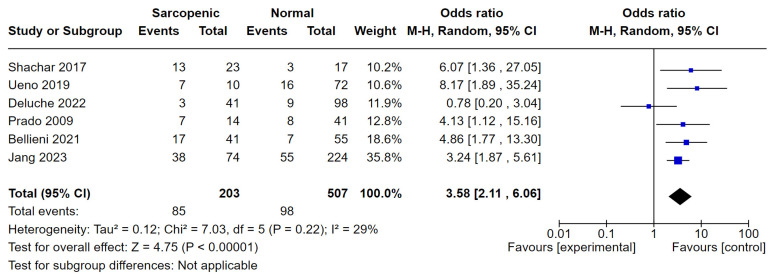
Forest plot analysis for drug-related toxicity and sarcopenia in breast cancer patients, Shachar 2017 et al. [15], Ueno 2019 et al. [28], Deluche 2022 et al. [31], Prado 2009 et al. [21], Bellieni 2021 et al. [29], Jang 2023 et al. [32].

**Table 1 cancers-16-00596-t001:** Characteristics of studies included in the meta-analysis.

First Author, Year of Publication	Type of the Study	Nation	Follow-Up Period	Patients (N), Disease Setting, and Breast Cancer Subtypes	Median (Range)/Mean (SD) Age, Years	Treatments	Sarcopenia Definition (cm^2^/m^2^)	Patients with Sarcopenia (N, %)	Measured Outcomes
Villasenor 2012 et al. [14]	Prospective	US	9.6 years	471, stage I–IIIA	56 (29.4–85.8) in sarcopenic and 63.9 (39.6–87.8) in non-sarcopenic patients	Adjuvant RT +/− CT/HT	LMI < 5.45 Kg/m^2^	75 (16)	5 y-OS, BCSS
Shachar 2017 et al. [15]	Prospective	US	1.9 years	40, MBC, 15 luminal, 14 HER2+, 10 triple negative	55 (34–80)	CT, TT	SMI ≤ 41	23 (48)	Toxicity OS
Caan 2018 et al. [16]	Retrospective	US	6.0 years	3241, stage II-III	54 (18–80)	Adjuvant RT +/− CT	SMI < 40	1086 (34)	mOS
Deluche 2018 et al. [17]	Retrospective	France	4.3 years	119 (88 ER+, 11 HER2+), stage I–IIIA	56 (21–87)	(Neo)adjuvant—CT +/− RT	SMI ≤ 41	58 (49)	DFS, OS
Hua 2020 et al. [18]	Retrospective	China	1.5 years	272 (197 ER+, 98 HER2+), stage I–III	45 (23–73)	Adjuvant-RT +/− CT	2 groups: low-SMI: 9.9 (range 5.3–10.6) and high-SMI: 12.5 (range 10.6–28.1)	66 (24)	OS, RFS
Jeon 2021 et al. [22]	Retrospective	Korea	6.5 years	479 (300 ER+), stage I–III, 237 luminal, 149 HER2 positive, 93 triple negative			SMI ≤ 41	178 (37)	OS, BCSS
Tang 2022 et al. [23]	Retrospective	China	5.2 years	97 (61 ER+, 40 HER2+), stage II–III			51 (21–87)	Adjuvant-CT	OS
Palleschi 2022 et al. [24]	Retrospective	Italy	2.7 years	25, stage I–III; 18, MBC. All HER2+.	58 (52–64)	CT +/− anti-HER2 agents	46 (27–73)	Adjuvant RT +/− CT	PFS, OS,
Ballinger 2023 et al. [25]	Prospective, phase III trial	US	Not specified (247 PFS events)	540 MBC, HR+ HER2−	63.2 (11.5)	HT+/− entinostat	SMI < 41	212 (39)	PFS, OS
Del Fabbro 2012 et al. [26]	Retrospective	US	7.7 years	129 (96 ER+, 44 HER2+), clinical stage I–III.	NA	Neoadjuvant-CT	SMI ≤ 38.5	18 (14)	pCR
Prado 2009 et al. [21]	Prospective	Canada	1 year	55 (39 ER+, 18 HER2+), MBC	54.8 (37–80)	CT	SMI ≤ 38.5	14 (26)	Toxicity, TTP
Ueno 2019 et al. [28]	Retrospective	Japan	4 years	82, clinical stage I–III.	55 (44.3–66)	Neoadjuvant -CT	SMI < 40	10 (12)	Toxicity
Bellieni 2021 et al. [29]	Retrospective	Italy	1 year	96, stage 0–III	77 (70–89)	Adjuvant-CT	LMI < 5.45 Kg/m^2^	41 (43)	Toxicity
Delrieu 2021 et al. [30]	Prospective	France	6 months	47 MBC	55 (10.41)	CT, TT, HT and RT	SMI < 40	25 (53)	Toxicity
Deluche 2022 et al. [31]	Prospective cross-sectional	France	6 months	139, MBC	61, 2(29.9–97.8)	CT, TT, HT	SMI < 39	41 (29)	Toxicity
Jang 2023 et al. [32]	Retrospective	Korea	5 months	298 stage I–III; luminal:103, HER2+:109, Triple negative:93	52.9 (overall SD NA)	Neoadjuvant CT	SMI ≤ 38.5	74 (25)	Toxicity (only the hematological)

Abbreviations: CT: chemotherapy, HT: hormone therapy, RT: radiotherapy, TT: target therapy, LMI: lean mass index, SMI: skeletal muscle index, BCSS: breast cancer-specific survival, DFS: disease-free survival, ER: estrogen receptor, HER2: human epidermal growth factor 2, OS: overall survival, PFS: progression-free survival, RFS: relapse-free survival, TTF: time to treatment failure, pCR: pathological complete response, NA: not available.

## Data Availability

The data presented in this study are available in this article.

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
