# Peer review of "Sarcopenia in Breast Cancer Patients: A Systematic Review and Meta-Analysis"

_cancers, 2024, doi:10.3390/cancers16030596_

Round 1
Reviewer 1 Report
Comments and Suggestions for Authors
In this study, the authors aimed to summarize current evidence about sarcopenia in non-metastatic as well as metastatic breast cancer (MBC), and to identify any correlation between sarcopenia and patients’ outcomes. The authors concluded that body composition is correlated with cancer and toxicity outcomes and suggest that breast cancer patients with low muscle mass in both metastatic and not-metastatic setting have more treatment related toxicity, and sarcopenia can be used to predict treatment related toxicity.
Comments
The reviewer has some concerns as follows:
1. What are the novel findings in this systematic review and meta-analysis? There seems to be only a statistical significance on the analysis for drugs-related toxicity and sarcopenia in breast cancer patients, which is in line with findings reported in a previous meta-analysis [ref. 26]. The authors should explain and discuss this issue.
2. In the Methods, lines 77-78, the authors described that “The literature search on Pubmed-MEDLINE was performed between July and October 2023”. There seems to be a mistake for “performed between July and October 2023”. Is there only 4 months searching?
3. In line 80, please check the round brackets “((((sarcopenia…”.
4. The legend for Figure 1 is lacking.
5. In Table 1, it needs to be re-organized. There are many incomplete words.
6. In line 248, please check “P = <0.0001”. Is it P < 0.0001 or P < 0.0001?
Author Response
Dear all,
thanks for your suggestions, we reported below point-by-point response to reviewers Independent Review Report, and changed the manuscript according as you can find in the “tracked” mode
Reviewer 1
Comments and Suggestions for Authors
In this study, the authors aimed to summarize current evidence about sarcopenia in non-metastatic as well as metastatic breast cancer (MBC), and to identify any correlation between sarcopenia and patients’ outcomes. The authors concluded that body composition is correlated with cancer and toxicity outcomes and suggest that breast cancer patients with low muscle mass in both metastatic and not-metastatic setting have more treatment related toxicity, and sarcopenia can be used to predict treatment related toxicity.
Comments
The reviewer has some concerns as follows:
- What are the novel findings in this systematic review and meta-analysis? There seems to be only a statistical significance on the analysis for drugs-related toxicity and sarcopenia in breast cancer patients, which is in line with findings reported in a previous meta-analysis [ref. 26]. The authors should explain and discuss this issue.
- In the Methods, lines 77-78, the authors described that “The literature search on Pubmed-MEDLINE was performed between July and October 2023”. There seems to be a mistake for “performed between July and October 2023”. Is there only 4 months searching?
- In line 80, please check the round brackets “((((sarcopenia…”.
- The legend for Figure 1 is lacking.
- In Table 1, it needs to be re-organized. There are many incomplete words.
- In line 248, please check “P = <0.0001”. Is it P < 0.0001 or P < 0.0001?
Author RESPONSE:
- Our metanalysis emphasized the role of sarcopenia as a predictor of higher rate of toxicity from oncological therapies in sarcopenic patients in both metastatic and non-metastatic breast patients, and especially more grade 3-4 toxicity in MBC. The last metanalysis published did not include a detailed analysis between the early and advanced breast cancer. We believe that our study adds more information as the previous metanalysis and reviews because we included even two studies of 2023 adding more information on the latest studies published. However, we stressed this concept in the discussion section as you have suggested.
- In the Methods, lines 77-78 “The literature search on Pubmed-Medline was performed between July and October 2023”- we intend that between July 2003 and October 2023 we systematically searched literature on Pubmed-Medline.
- The round blankets were checked in line 80 as suggested.
- We included the legend for Figure 1 as indicated.
- We re-organized Table 1 with all the missing words
- In line 248 P <0.00001
Reviewer 2 Report
Comments and Suggestions for Authors
1. The abstract is not formatted according to the rules: it is necessary to use subheadings and numbering.
2. The abstract states that the study was carried out during the period July - October 2023, but it is better to indicate that the literature search was for the period N years / up to October 2023 inclusive or something else.
3. In Figures 3 and 4, what does the subtitle “New subgroup” mean?
4. Was sarcopenia related to tumor size, lymph node involvement, or patient age? The authors considered only cases of metastatic and non-metastatic breast cancer, why?
5. Was sarcopenia unrelated to the treatment regimen? Have you checked this question?
In general, the work is quite interesting, well presented, and undoubtedly deserves the attention of readers. However, I have the impression that the data obtained from 16 sources can be analyzed in more depth.
Author Response
Dear all,
thanks for your suggestions, we reported below point-by-point response to reviewers Independent Review Report, and changed the manuscript according as you can find in the “tracked” mode
Reviewer 2
Comments and Suggestions for Authors
- The abstract is not formatted according to the rules: it is necessary to use subheadings and numbering.
- The abstract states that the study was carried out during the period July - October 2023, but it is better to indicate that the literature search was for the period N years / up to October 2023 inclusive or something else.
- In Figures 3 and 4, what does the subtitle “New subgroup” mean?
- Was sarcopenia related to tumor size, lymph node involvement, or patient age? The authors considered only cases of metastatic and non-metastatic breast cancer, why?
- Was sarcopenia unrelated to the treatment regimen? Have you checked this question?
In general, the work is quite interesting, well presented, and undoubtedly deserves the attention of readers. However, I have the impression that the data obtained from 16 sources can be analyzed in more depth.
Author RESPONSE:
- We formatted the abstract as suggested using subheadings and numbering.
- We corrected the abstract statement about the period when the study was carried out as suggested.
- In figure 3 and 4 the subtitle New Subgroup dos does not mean anything and is an error
- It is not possible to obtain subgroup analyzes based on the clinical pathological characteristics of the patients from the studies that we analyzed. It was possible from the studies included to evidence a difference only based on the setting of the disease: early vs. metastatic breast cancer.
- It is not possible to extrapolate this information on the difference based on the treatment regimen since the studies are very heterogeneous. In any case, the type of treatment carried out is shown in table 1
Reviewer 3 Report
Comments and Suggestions for Authors
This is an interesting systematic review and meta analysis with adequate quality and novelty. Some points should be addressed.
- Abstract: Some abbreviations should be explained, e.g., CT, MBC, DXA.
- Introduction, lines 44-49: The first 3 sentences should be incorporated into one paragraph. In this first paragraph, some more information about the epidemiological data of breast cancer (metastatic and non-metastatic) should be added and especially concerning data of the country from which the patients are belonged.
- Introduction, lines 57-60: These statements need a bit higher explanation.
- Introduction, lines 61-63: This statement needs much more explanation in order to be a full paragraph. For example, what type of survival is correlated with sarcopenia in these types of cancers? What about the kind of therapy used in these types of cancers?
- Introduction, line 68: The authors reported that sarcopenia is related with quality of life of breast cancer patients. This statement should be enriched with further information due to its major significance.
- Materials and Methods, line 86: Please revise "man" to "men".
- Materials and Methods: Concerning selection criteria, were only English language written studies included in meta-analysis? Please, add a statement for the above.
- Materials and Methods: The last paragraph of 2.2 section could be split into two paragraphs, maybe at line 109.
- A figure legend for Figure 1 is missing.
- Lines 200-201: The statement "With a follow-up range among the studies of 6 months up to >6 years". This statement is a bit confusing and needs rephrasing.
- Table 1: What are Pcr, TTP, ER, HER2? These should be included in the abbreviations of the table.
- Discussion, 1st paragraph: The authors reported that there are two previous meta analysis studies in breast cancer. In this point, more information about these meta-analysis should be added. Moreover, a comparison of the previous meta-analysis studies with the present study shlould be performed and described in order to emphasize the novelty of the present meta-analysis.
- Discussion, lines 262-265: This sentence is a bit confusing and it needs to be rephrased and revised concerning its syntax.
- Discussion, lines 272-273: The authors reported that "... sarcopenia has recently emerged as a new condition, that independently from malnutrition". Firstly, "that indepedently from malnutrition" is confusing. Do you mean ".... a new condition, indepedently of malnutrition"?
- Moreover, this issue about the association of sarcopenia with malnutrition should be more described due to its high significance by reporting relevant references.
- Discussion, lines 284-285: The authors report that "Interestingly, in Aleixo work, low muscle density was prognostic for OS in MBC but not in the early stage." This sentence needs revision about its syntax, e.g., "... low muscle density was identified as a progrnostic factor for ....". Why the above its interesting? The authors should explain at this point the reasons for the above.
- Discussion, lines 288-290: Thise sentence is a bit confusing and needs rephrasing.
- Discussion, lines 287-299: These two paragraphs should be merged into one paragraph.
- Discussion, line 304. A dot should be added after "with caution". Then, a new sentence should begin, e.f. "Firstly, ...".
- Discussion, lines 313-322: These two paragraphs should be merged into one paragraph.
- Discussion, lines 320-322: This statement could be enriched with a bit more information due to the significance of the nutritional status of breast cancer patients.
- Conclusion, line 345: "... and we hypothesise could prevent severe ...". The above statement needs revision in order to not be confusing.
Comments on the Quality of English LanguageModerate editing of English language is required.
Author Response
Dear all,
thanks for your suggestions, we reported below point-by-point response to reviewers Independent Review Report, and changed the manuscript according as you can find in the “tracked” mode
Reviewer 3
Comments and Suggestions for Authors
This is an interesting systematic review and meta analysis with adequate quality and novelty. Some points should be addressed.
- Abstract: Some abbreviations should be explained, e.g., CT, MBC, DXA.
- Introduction, lines 44-49: The first 3 sentences should be incorporated into one paragraph. In this first paragraph, some more information about the epidemiological data of breast cancer (metastatic and non-metastatic) should be added and especially concerning data of the country from which the patients are belonged.
- Introduction, lines 57-60: These statements need a bit higher explanation.
- Introduction, lines 61-63: This statement needs much more explanation in order to be a full paragraph. For example, what type of survival is correlated with sarcopenia in these types of cancers? What about the kind of therapy used in these types of cancers?
- Introduction, line 68: The authors reported that sarcopenia is related with quality of life of breast cancer patients. This statement should be enriched with further information due to its major significance.
- Materials and Methods, line 86: Please revise "man" to "men".
- Materials and Methods: Concerning selection criteria, were only English language written studies included in meta-analysis? Please, add a statement for the above.
- Materials and Methods: The last paragraph of 2.2 section could be split into two paragraphs, maybe at line 109.
- A figure legend for Figure 1 is missing.
- Lines 200-201: The statement "With a follow-up range among the studies of 6 months up to >6 years". This statement is a bit confusing and needs rephrasing.
- Table 1: What are Pcr, TTP, ER, HER2? These should be included in the abbreviations of the table.
- Discussion, 1st paragraph: The authors reported that there are two previous meta analysis studies in breast cancer. In this point, more information about these meta-analysis should be added. Moreover, a comparison of the previous meta-analysis studies with the present study shlould be performed and described in order to emphasize the novelty of the present meta-analysis.
- Discussion, lines 262-265: This sentence is a bit confusing and it needs to be rephrased and revised concerning its syntax.
- Discussion, lines 272-273: The authors reported that "... sarcopenia has recently emerged as a new condition, that independently from malnutrition". Firstly, "that indepedently from malnutrition" is confusing. Do you mean ".... a new condition, indepedently of malnutrition"?
- Moreover, this issue about the association of sarcopenia with malnutrition should be more described due to its high significance by reporting relevant references.
- Discussion, lines 284-285: The authors report that "Interestingly, in Aleixo work, low muscle density was prognostic for OS in MBC but not in the early stage." This sentence needs revision about its syntax, e.g., "... low muscle density was identified as a progrnostic factor for ....". Why the above its interesting? The authors should explain at this point the reasons for the above.
- Discussion, lines 288-290: Thise sentence is a bit confusing and needs rephrasing.
- Discussion, lines 287-299: These two paragraphs should be merged into one paragraph.
- Discussion, line 304. A dot should be added after "with caution". Then, a new sentence should begin, e.f. "Firstly, ...".
- Discussion, lines 313-322: These two paragraphs should be merged into one paragraph.
- Discussion, lines 320-322: This statement could be enriched with a bit more information due to the significance of the nutritional status of breast cancer patients.
- Conclusion, line 345: "... and we hypothesise could prevent severe ...". The above statement needs revision in order to not be confusing.
Author RESPONSE
- We explained the abbreviations in the Abstract
- We reviewed the paragraph as suggested. Unfortunately, we don’t have the last epidemiological data of the patients in the studies considered. For that reason, we didn’t include these data n our review
- We modified the lines as suggested
- We explained better the lines
- We enriched the statement in line 68
- We corrected that as indicated
- We added the statement in the text
- We corrected the statement
- A figure legend for Figure 1 was added
- Lines 200-201 were rephrased
- We added the explanation missing
- Our metanalysis emphasized the role of sarcopenia as a predictor of higher rate of toxicity from oncological therapies in sarcopenic patients in both metastatic and non-metastatic breast patients, and especially more grade 3-4 toxicity in MBC. The last metanalysis published did not include a detailed analysis between the early and advanced breast cancer. We believe that our study adds more information as the previous metanalysis and reviews because we included even two studies of 2023 adding more information on the latest studies published.
- The lines 262-265 were rephrased
- - Discussion, lines 272-273 were modified
- we have added in thew discussion section the following refereces : 40. Beaudart C, Sanchez-Rodriguez D, Locquet M, Reginster JY, Lengelé L, Bruyère O. Malnutrition as a Strong Predictor of the Onset of Sarcopenia. Nutrients. 2019 Nov 27;11(12):2883. DOI: 10.3390/nu11122883.
- Su CH, Chen WM, Chen MC, et al. The Impact of Sarcopenia Onset Prior to Cancer Diagnosis on Cancer Survival: A National Population-Based Cohort Study Using Propensity Score Matching. Nutrients. 2023 Mar 1;15(5):1247. DOI: 10.3390/nu15051247
- Muscaritoli M, Bar-Sela G, Battisti NML, et al. Oncology-Led Early Identification of Nutritional Risk: A Pragmatic, Evi-dence-Based Protocol (PRONTO). Cancers (Basel). 2023 Jan 6;15(2):380. DOI: 10.3390/cancers15020380
- Muscaritoli M, Modena A, Valerio M, Marchetti P, Magarotto R, Quadrini S, Narducci F, Tonini G, Grassani T, Cavanna L, Di Nunzio C, Citterio C, Occelli M, Strippoli A, Chiurazzi B, Frassoldati A, Altavilla G, Lucenti A, Nicolis F, Gori S. The Im-pact of NUTRItional Status at First Medical Oncology Visit on Clinical Outcomes: The NUTRIONCO Study. Cancers (Basel). 2023 Jun 15;15(12):3206. DOI: 10.3390/cancers15123206.
- Discussion, lines 284-285: were explained as indicated.
- Discussion, lines 288-290: were explained.
- Discussion, lines 287-299: were modified as suggested.
- Discussion, line 304 was modified.
- These two paragraphs should be merged into one paragraph.
- Discussion, lines 320-322: the statement was enriched.
- Conclusion, line 345: were rephrased.
Round 2
Reviewer 1 Report
Comments and Suggestions for Authors
This revised manuscript has a great improvement and can be accepted.
Author Response
thanks for your suggestion and revision
Reviewer 2 Report
Comments and Suggestions for Authors
The authors responded to the reviewer's comments. Please note that in line 106 - between July and October 2023 will skip a year from July (2003?).
Author Response
thanks for your revision. we have changed in July 2023
Reviewer 3 Report
Comments and Suggestions for Authors
The authors have significantly improved their manuscript, reinforcing its aim and the literature gap that they effectivelly covered, as well as their results and conclusions.
Comments on the Quality of English LanguageMinor editing of English language required
Author Response
Thanks for your revision. we have done some editing of language as your suggestion